# Effects of long-term plate fixation with different fixation modes on the radial cortical bone in dogs

**Norihiro Muroi**[1]*, **Hiroki Ochi**[2], **Masakazu Shimada**[1], **Yoshinori Asou**[3], **Yasushi Hara**[1]

**1** Department of Veterinary Surgery, Nippon Veterinary and Life Science University, Musashino-shi, Tokyo, Japan, **2** Department of Rehabilitation for Movement Functions, Research Institute, National Rehabilitation Center for Persons with Disabilities, Tokorozawa-shi, Saitama, Japan, **3** Department of Nano-Medicine, Graduate school of Medical and Dental Sciences, Tokyo Medical and Dental University, Bunkyo-ku, Tokyo, Japan

* fishertiger.sun@gmail.com

**Data Availability Statement:** All relevant data are within the manuscript and its Supporting Information files.

**Funding:** The author(s) received no specific funding for this work.

## Abstract

The aim of this study was to examine the effect of long-term locking plate fixation on the cortical bone of the canine radius. Locking compression plates were fixed to the left and right radius in dogs (n = 3). The left radius was fixed with a locking head screw (Locking Plate group, LP). The locking compression plate was compressed periosteally in the right radius using a cortex screw (Compression Plate group, CP). Radial bones from dogs that were euthanized for other purposes were collected as an untreated control group (Control group). After euthanasia at 36 weeks following plate fixation, radial bones were evaluated for bone mineral density and underwent histological analysis. Bone metabolic markers were analyzed by quantitative polymerase chain reaction (qPCR). Statistical analyses were performed for comparisons between groups. The LP group showed no significant difference in bone mineral density after plate fixation, whereas the CP group showed significantly lower bone mineral density. Histological analysis indicated that the number of osteoclasts and rate of empty lacunae increased significantly in the CP group relative to the Control and LP groups. qPCR analysis indicated increased expression of inflammatory cytokines, such as *tumor necrosis factor-alpha*, *interleukin-6*, and *tumor necrosis factor ligand superfamily member 11* in the CP group, whereas *Runt-related transcription factor 2*, an osteoblast marker, was similar in all groups. The expression of *hypoxia-inducible factor-1α* in the CP group was also increased relative to that in the Control and LP groups. Thus, locking plate fixation is a biologically superior fixation method that does not cause implant-induced osteoporosis in the bone in the long term.

## Introduction

In cases of small animal orthopedic surgery, internal fixation with conventional plates and screws is a reliable treatment method for long bone fractures [1, 2]. However, application of conventional plates and screws induces structural changes in the bone, with cortical

**Competing interests:** The authors have declared that no competing interests exist.

osteonecrosis of the bone occurring just below the plate and cortical bone thinning of about 40% occurring at 24 weeks after dynamic compression plate placement [3], known as implant-induced osteoporosis (IIO). IIO causes biphasic changes due to inadequate blood supply at 8–12 weeks and reduced mechanical stress on the bones at 24–36 weeks [4]. IIO is relatively common in small dogs, especially those with insufficiently developed bone microvessels [5], after internal fixation with a conventional plate [6]. This phenomenon predisposes to re-fractures after implant removal [7].

Locking plates that preserve blood flow to the periosteum and enable angularly stable fixation have recently been introduced to the small animal clinical field [8, 9]. In contrast to conventional plates, which provide stability between bone fragments by frictional forces between the plate and bone, locking plates allow the plate to be placed away from the periosteal surface and do not require compression of the periosteum, preserving periosteal blood flow and achieving secondary bone healing due to relative stability. Preserving periosteal blood flow during fracture treatment is an important aspect for fracture healing. Appropriate blood flow reduces the risk of infection and IIO. Locking plates with a small periosteal contact area reportedly reduce the risk of early postoperative osteoporosis [10]. However, studies have not yet clarified whether the locking plate system, which should be biologically superior to the conventional plate system, has better clinical outcomes than the conventional plate system [11]. In addition, the histological and molecular biological findings for a comparison of the long-term effects of the locking and conventional systems on bone have not been identified.

It has been reported that there is a correlation between plate contact width and porosity in cortical bone necrosis beneath the plate that occurs after plate placement [12]. The bone loss in this phenomenon is induced by necrotic bone which stimulates internal remodeling of the Haversian system. In addition, the plate placement has been shown to impair the cortical bone blood flow beneath the plate [13]. Internal remodeling involves the following steps: bone resorption by osteoclasts, followed by bone formation by osteoblasts, and finally osteocyte connection and blood supply. Bone resorption capacity can be predicted by observing RANKL (gene: *Tnfsf11*), which induces osteoclast differentiation, and the number of osteoclasts. In addition, osteogenic capacity can be predicted by observing RUNX2 [14], which is essential for osteoblast differentiation. By comparing bone resorption and bone formation, the metabolic conditions of bone can be assessed. Furthermore, the degree of blood flow impairment can be predicted by observing the vascular endothelial marker, CD31 [15]. By observing the pro-inflammatory cytokines TNF-α and IL-6 [16], and hypoxia-inducible factor 1-alpha (HIF1-α) [17], we can evaluate the enhanced expression of RANKL due to inflammation and hypoxia in bone caused by impaired blood flow.

This study aimed to compare the effects of 36 weeks of plate fixation on the radius in a fixation model with periosteal compression (impaired blood flow model) and in a fixation model without periosteal compression (non-impaired blood flow model).

## Materials and methods

### Animals

TOYO Beagle dogs were purchased from KITAYAMA LABES CO., LTD (Nagano, Japan). For the experimental group, unspayed healthy female beagle dogs aged approximately 1 year (n = 3; weight, 9.6 ± 0.6 kg) were used. Blood tests, radiographs, and Computed tomography (CT) scans were performed prior to the surgical procedure to ensure that the animals had no physical abnormalities. CT scans were taken of the entire body, including the bones of the arms and legs, spine, and organs. In addition, the radial bones harvested from the dogs euthanized for other reasons (approximately 1 year old; unspayed females; weight, 9.9 ± 1.0 kg;

n = 10) were used as an untreated control group (Control group). There was no history of the use of drugs that affected bone metabolism in either group. In the control group, eight samples underwent histological examination and gene expression analysis while the remaining two samples underwent only gene expression analysis. This experiment was approved by the Nippon Veterinary and Life Science University's Laboratory Animal Ethics Committee (approval number: 29S-16).

## Surgical procedure

Midazolam hydrochloride (0.2 mg/kg, i.v.) and buprenorphine (0.02 mg/kg, bid, i.v.) were administered as preanesthetic agents. Propofol (7 mg/kg, i.v.) was used to induce anesthesia, and isoflurane was used to maintain anesthesia. In addition, local anesthesia (RUMM block, bupivacaine hydrochloride: 1 mg/kg) was used for analgesia. Antibiotics (cefazolin sodium, 20 mg/kg, bid, i.v.) were administered preoperatively and intraoperatively (every 2 hours).

The left and right forearms were shaved and surgically disinfected. After making an incision lateral of the cephalic vein from the radial head to the distal bone end, the radial metaphysis was exposed by incising the fascia between the extensor carpi radialis and the common digital extensor muscles. The abductor pollicis longus muscle was not resected to preserve as much of the soft tissue as possible. However, part of the pronator muscle was dissected because it was an obstacle to plate placement. The midpoint of the radius was identified from the distal radius end and was used as the plate (midpoint) placement point. Locking compression plates (LCP) of the same standard (length, 81 mm; thickness, 2.6 mm; width, 7.5 mm; nine holes, Code No. VP4031-09; DePuy Synthes Companies, Raynham, MA, U.S.A.) were used. LCP contouring was performed with reference to preoperative CT images for the left radius (LP group) and the locking head screw (diameter, 2.7 mm; length, 12 mm; code No. VS206-12; DePuy Synthes Companies) was fixed to the LCP using a 0.8-N·m torque limiter (Code No. 511–776; DePuy Synthes Companies). At this time, the LCP was fixed with a spacer (stainless steel, 3-mm-thick) between the LCP and the bone, and a gap of 3 mm between the LCP and the bone surface was secured. The spacer was removed after the locking head screw was fixed. Of the nine holes in the LCP, the locking head screws were inserted and fixed in three holes at the most distal and the proximal side of each; the middle three holes were left empty. For the right radius (CP group), the LCP was fixed on the cranial surface of the radius by the cortex screw (diameter, 2.7 mm; length, 12 mm; code No. 51-102-12; Mizuho Ikakogyo, Tokyo, Japan) using a 1.5-N·m torque limiter (Code No. 511–115; DePuy Synthes Companies). Of the nine holes in the LCP, the cortex screws were inserted and fixed in three holes at the most distal and the proximal side of each; the middle three holes were empty.

The Robert Jones bandage technique was performed for 2 days to protect the surgical wound and reduce inflammation and edema, and the dogs received an antibiotic (cefalexin, 20 mg/kg, bid, p.o.) and an analgesic (buprenorphine, 0.02 mg/kg, bid, s.c.) for 2 weeks. The sutures were removed 2 weeks after the procedure.

The dogs were kept in a cage (length, 1.15 m; width, 0.7 m; and height, 1.5 m) under free-drinking, twice-daily feeding (general diet), long days, and a room temperature of 25˚C. After 36 weeks of plate and screw placement, the dogs were euthanized with anesthetics (Pentobarbital, 100 mg/kg, i.v.).

## Radiographic examination

Anterior-posterior (46 kV, 5 mAs) and lateral (46 kV, 5 mAs) radiographs of the forearm were taken preoperatively, immediately after surgery, and at 4-week intervals from 4 to 36 weeks postoperatively using an X-ray machine (VPX-200; TOSHIBA, Tokyo, Japan) to assess implant

loosening or failure, and to evaluate the radiographic density of the radial bone just below the plate.

## Quantitative computed tomography (QCT)

Preoperative and post-euthanasia CT scans (TSX-303A; TOSHIBA) were performed (slice width, 0.5 mm; slice interval, 0.5 mm; tube voltage, 120 kV; tube current, 300 mA). Forearm and CT bone mineral density phantoms (B-MAS200; Kyotokagaku, Kyoto, Japan) were taken simultaneously. Osirix MD (ver. 10.0.2) was used to extract the volume (including the cortical and medullary cavity) of the region of interest (center of the plate: midway between the third and fourth screws) from the axial section, and bone mineral density was estimated from the average CT values using the QCT method (Fig 1) [18].

## Preparation of tissue sections

The bones of the radius between the third and fourth screws (center of the plate) was harvested, and divided into 1-cm segments using a diamond cutter. The segmented bones were fixed in 4% PFA at 4˚C for 2 days and then demineralized with 10% EDTA at 4˚C for 8 weeks. After demineralization was completed, paraffin embedding was performed. After embedding, thin paraffin sections of 1.5–4.5 μm were prepared using a microtome.

## Histology and immunohistochemistry

**Hematoxylin and eosin (HE) staining.** Paraffin sections were deparaffinized with xylene and then hydrated with 99.5% ethanol. The sections were washed with distilled water and soaked in hematoxylin staining solution for five min. Then, they were washed again with distilled water and soaked in water-soluble eosin staining solution for five min. Sections were dehydrated with 90% ethanol (3 times, 5 min each) and 99.5% ethanol (2 times, 5 min each), in that order. Permeation was performed with xylene. After sealing with Marinol and cover glass, observations were performed under a microscope. The cortical and medullary cavity areas (calculated as a percentage of the total area) and empty lacunae rate (average of the

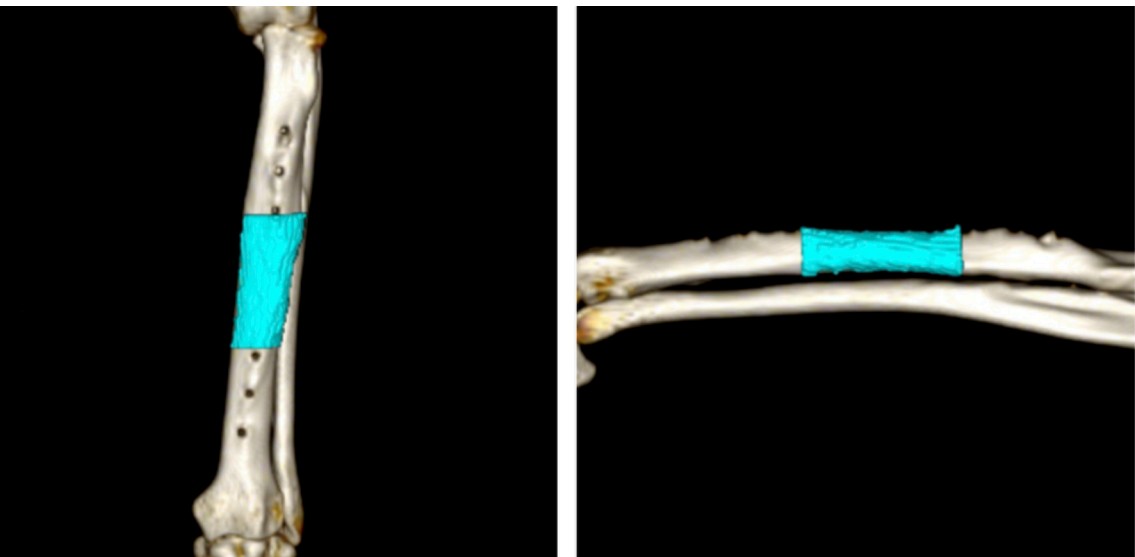

**Fig 1. Measurement areas of bone mineral density and volume in the radius on computed tomography images.**

percentage of empty lacunae in the osteocytes out of 50 in five fields of view at 200x magnification) in the cranial, middle, and caudal areas of the transverse radial section determined by HE staining were obtained using ImageJ (ver. 1.51s) image processing software (Fig 2).

**Tartrate-resistant acid phosphatase (TRAP) staining.** Deparaffinization, hydration, and washing were performed as described above, followed by immersion in the blocking solution (a mixture of 0.2 M acetate buffer, sodium tartrate, and ion exchange water) for 10 min at room temperature. After immersion in the substrate reaction solution (0.2 M acetate buffer, sodium tartrate, 1 N sodium hydroxide, naphthol AS-BI phosphate, pararose aniline, hydrogen chloride, sodium nitrite and ion exchange water) for 20 min at 37˚C, washing with distilled water and nuclear staining with hematoxylin staining solution were performed. Dehydration, permeability, and encapsulation were performed as described above. TRAP staining was used to count the number of osteoclasts in the cranial, middle, and caudal areas of the radial transverse section (the number in each area was divided by the cortical bone area).

**Immunostaining.** After deparaffinization, hydration, and washing as described above, endogenous peroxidase was inhibited by incubation in 30% hydrogen peroxide in methanol (room temperature, 30 minutes). After washing with PBS, antigen inactivation treatment (65˚C, 60 min) was performed with citrate buffer. After washing in PBS and blocking with Block Ace (KAC, Kyoto, Japan) (room temperature, 30 minutes), samples were incubated (room temperature, overnight) with primary antibody (Anti-RUNX2 antibody: Code No. ab23981; Abcam, Cambridge, UK; Anti-CD31 antibody: Code No. bs-0468R; Bioss Antibodies Inc., Woburn, MA, U.S.A.). After washing with PBS, samples were incubated (room temperature, 60 minutes) with secondary antibody. Next, after washing in PBS, coloration was performed using diaminobenzidine under a microscope, and the reaction was stopped at the appropriate point. After washing with distilled water, nuclear staining was performed with hematoxylin staining solution. Dehydration, permeability, and encapsulation were performed in the same manner as described above. The number of RUNX2-positive osteoblasts and CD31-positive vascular endothelial cells was counted in the cranial, middle, and caudal areas of the radial transverse section (the number in each area was divided by the area of cortical bone).

## RNA extraction and quantitative polymerase chain reaction (qPCR)

The harvested radius was crushed using liquid nitrogen and a bead crusher. RNA extraction was performed by the AGPC method. The procedure is as follows: The crushed radius (1,000

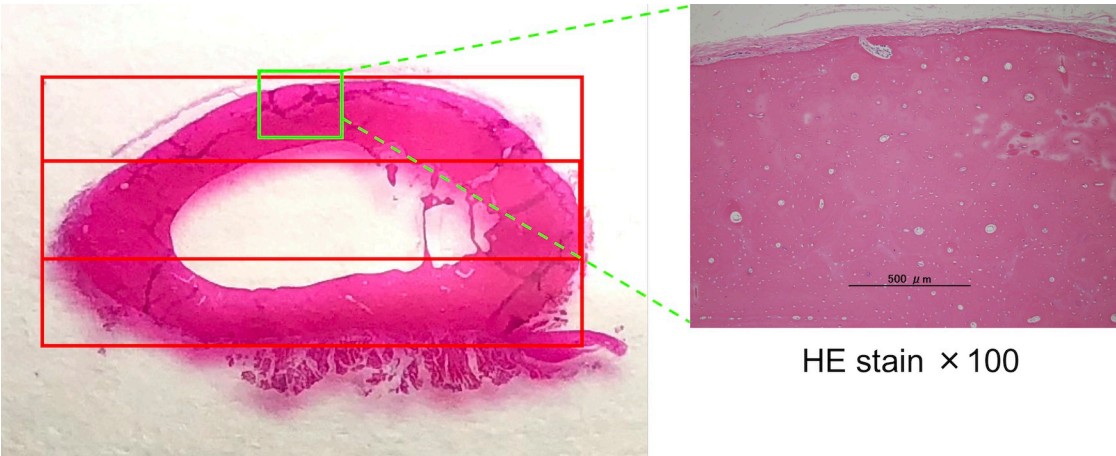

HE stain × 100

**Fig 2. Division of the cross-section of the radius during measurement.** The transverse section of the radius shows three areas. The upper part is the cranial region, the middle part is termed the middle region, and the lower part is the caudal region.

**Table 1. Primer sequences for real-time polymerase chain reaction.**

| Gene name | Gene symbol; | Ref. sequence | Primer |
|---|---|---|---|
| Actin, beta | *ACTB* | XM_005621019.3 | Forward 5'–AGGAAGGAAGGCTGGAAGAG–3' |
| | | | Reverse 5'–TGCGTGACATCAAGGAGAAG–3' |
| Tumor necrosis factor ligand superfamily member 11 | *Tnfsf11* | XM_846672.4 | Forward 5'–GCACCAAATACTGGTCAGGGA–3' |
| | | | Reverse 5'–TGGGTCCAGTAGTGATGGGT–3' |
| Runt-related transcription factor 2 | *RUNX2* | XM_022425793.1 | Forward 5'– CAGACCAGCAGCACTCCATA–3' |
| | | | Reverse 5'–CAGCGTCAACACCATCATTC–3' |
| Cluster of differentiation 31 | *CD31* | XM_022422844.1 | Forward 5'–CACCTGAGCCTTACCAAGAGAA–3' |
| | | | Reverse 5'–GGGATGTGATGTCCTCTCAGG–3' |
| Tumor necrosis factor alpha | *TNF-α* | NM_001003244.4 | Forward 5'–TCATGTTGTAGCAAACCCCGA–3' |
| | | | Reverse 5'–TACAACCCATCTGACGGCAC–3' |
| Interleukin 6 | *IL-6* | NM_001003301.1 | Forward 5'–GTGCACATGAGTACCAAGATCC–3' |
| | | | Reverse 5'–GTCTGTGGTTGGGTCAGGAG–3' |
| *Hypoxia-inducible factor 1 alpha* | *HIF-1α* | NM_001287163.1 | Forward 5'–TGACGGTTCACTTTTTCAAGC–3' |
| | | | Reverse 5'–TTGCTCCATTCCATTCTGTTC–3' |

Dog-specific primers were designed using NCBI Primer-BLAST.

mg) was immersed in 1,000 μL of TRIZOL® Reagent (Invitrogen, CA, U.S.A.). After incubation, a mixture containing 0.2 mL of chloroform per mL of TRIZOL® Reagent was added and mixed vigorously. After incubation, centrifugation was performed to separate the RNA phase. The RNA phase was transferred to a separate tube, a mixture containing 0.25 mL of isopropanol and high salt buffer per mL of TRIZOL® Reagent was added and mixed. After incubation, RNA was precipitated and pelleted by centrifugation. After removing the supernatant following centrifugation, the RNA pellet was washed with 75% ethanol. Next, a mixture containing 1 mL of 75% ethanol per mL of TRIZOL® Reagent was added and centrifuged after vortexing. After removing the supernatant, the pellet was dried, and RNA was dissolved in RNase-free water. The concentration and quality of RNA was evaluated using a NanoDrop. RNA and Syber green (One Step TB Green® Prime Script™ PLUS RT-PCR Kit; Takara Bio Inc., Shiga, Japan) were used to perform qPCR reactions. To evaluate bone metabolism, we measured the expression of *Tnfsf11* (The gene encoding RANKL), which is involved in differentiation into mature osteoclasts as bone resorption activity, and *RUNX2*, which is involved in differentiation into mature osteoblasts as bone formation activity. In addition, we measured the expression of *CD31* as a vascular endothelial cell marker to evaluate vessel count, and the expression of *TNF-α* and *IL-6* (pro-inflammatory cytokines), which affect RANKL expression, to evaluate local inflammation. The expression of *HIF-1α* (hypoxia-inducible factor 1-alpha) was quantified to assess hypoxia due to reduced periosteal blood flow or inflammation caused by periosteal compression. The relative mRNA expression levels of *Tnfsf11*, *RUNX2*, *CD31*, *TNF-α*, *IL-6*, and *HIF-1α* were calculated using the ΔΔCT method with the *ACTB* housekeeping gene as an internal control. The primers for each gene are listed in Table 1.

## Statistical analysis

The untreated control group was defined as the Control group, the group in which LCPs were fixed with locking head screws was defined as the Locking Plate (LP) group, and the group in which LCPs were fixed with cortex screws was defined as the Compression Plate (CP) group. Statistical analysis (Tukey-Kramer test) between each group was performed using R (ver. 3.6.2). A p value of $< 0.05$ was considered statistically significant.

## Results

### Postoperative status

Heat and swelling of the operative wound were observed for 2 days postoperatively, but no similar symptoms were observed thereafter in any of the dogs (n = 3) until euthanasia. Pain by palpation and lameness in forearms were noted for 1 week postoperatively, but there was no obvious pain and lameness thereafter. There was no heat, edema, pain, or secretion of serous fluid from the operative wound 2 weeks postoperatively. The stitches were removed. There were no complications such as interdigital dermatitis.

### Radiographic evaluation

There was no evidence of implant loosening or failure across 36 weeks in either group (Fig 3). In the LP group, there was no evidence of increased radiolucency in the cortical bone just below the plate. However, periosteal bone proliferation at the screw insertion site proximal to the plate and decreased radiolucency of the bone marrow cavity were observed. In contrast, thinning of the cranial cortical bone just below the plate in the CP group was observed after 12 weeks. There was decreased radiolucency of the bone marrow cavity at the screw insertion site proximal to the plate in the CP group.

### QCT method for evaluation of bone mineral density and volume

In the LP group, the bone mineral density during 36 weeks of implantation was not significantly different from the value before implantation (pre-, 837.2 mg/cm$^3$; post-, 804.7 mg/cm$^3$). In contrast, in the CP group, the bone mineral density significantly decreased during 36 weeks of implantation (pre-, 834.3 mg/cm$^3$; post-, 759.2 mg/cm$^3$, p = 0.035). There was no significant

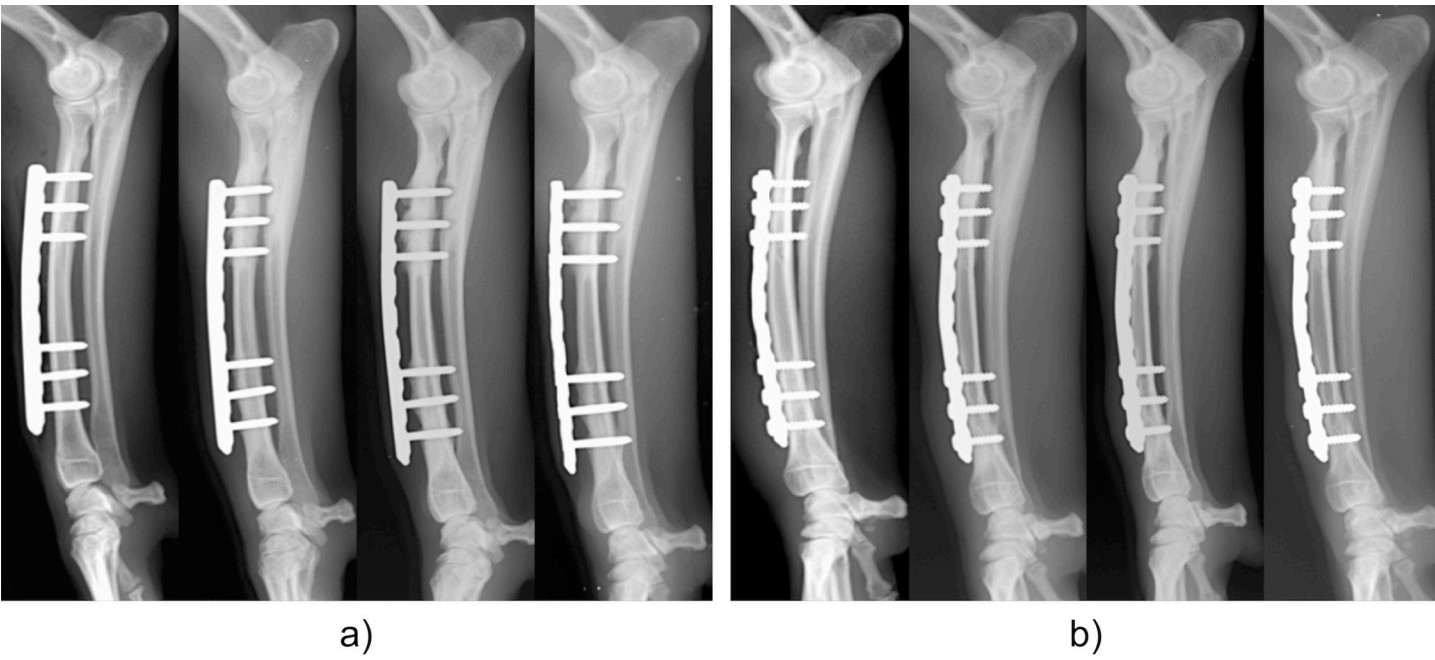

a)                                                                    b)

**Fig 3. Continuous changes observed on X-rays.** a) LP group. b) CP group. Left to right: Immediately after implant placement, and 12, 24, and 36 weeks after surgery in each group. Both the LP and CP groups show increased opacity of the bone area around the screw proximal to the plate. No implant loosening or failure can be observed.

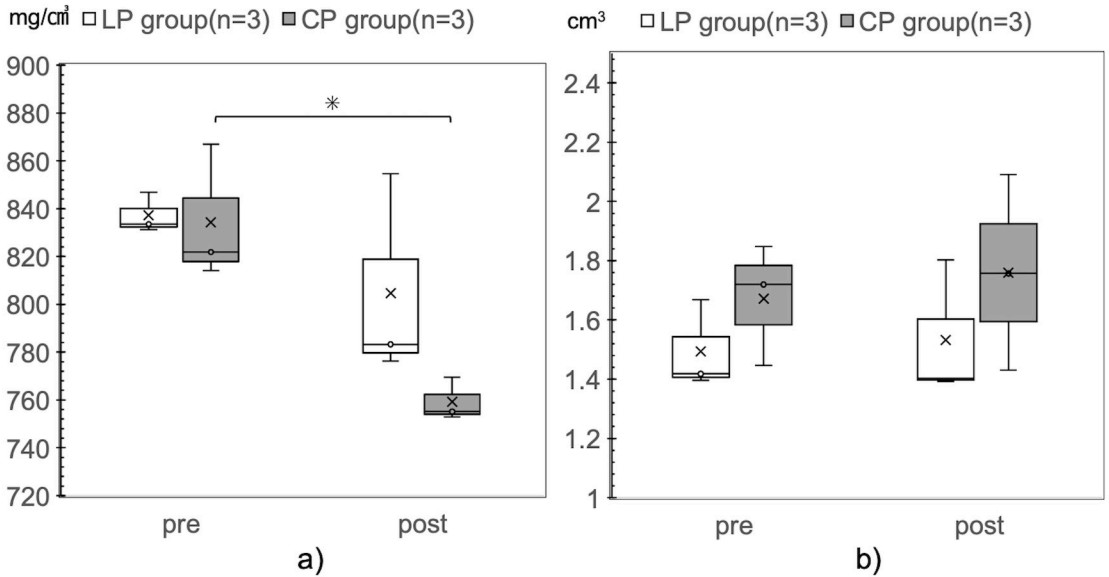

**Fig 4. Comparison of bone mineral density and bone volume before implant placement and after euthanasia.** a) Bone mineral density. b) Bone volume. The CP group shows a significant decrease in bone mineral density, relative to that in the pre-implant period, 36 weeks after placement of the implants (pre, 834.3 mg/cm$^3$; post, 759.2 mg/cm$^3$; $^*$p = 0.035).

difference in bone mineral density between the LP and CP groups preoperatively and 36 weeks after surgery (Fig 4A).

There was no significant difference in bone volume before and 36 weeks after implant placement in either group (Fig 4B).

## Morphologic evaluation of radius cross-section

There was no significant difference in the area ratio of the cortical and medullary areas between the groups in the cranial, middle, and caudal areas. However, the area ratio of the cranial area of the radial cortical bone was lower in the CP group than in the Control group (Control group, 0.23; CP group, 0.2; p = 0.06) (Fig 5A, Table 2).

## Comparison of empty lacunae rate and bone metabolism markers in radius cross-section

Compared with the Control and LP groups, the CP group showed a significant increase in the empty lacunae rate (Control group, 0.20; LP group, 0.24; CP group, 0.39; Control vs CP group, p = 0.005; LP vs CP group, p = 0.047) and osteoclast numbers (Control group, 0.05 cells/mm$^2$; LP group, 0.17 cells/mm$^2$; CP group, 1.03 cells/mm$^2$; Control vs CP group, p = 0.002; LP vs CP group, p = 0.016) in the cranial area (Fig 5, Table 3). In the middle area and whole bone area, the CP group tended to show an increased number of osteoclasts compared with those in the Control and LP groups. There was no significant intergroup difference in osteoclast numbers in the caudal area.

Compared with the Control and LP groups, the CP group showed a trend for increased osteoblasts (control group, 15.09 cells/mm$^2$; LP group, 18.55 cells/mm$^2$; CP group, 25.44 cells /mm$^2$; control vs CP, p = 0.08; LP vs CP, p = 0.33) in the cranial, middle, and caudal areas as well as the entire bone. There was no significant intergroup difference in the number of CD31-positive cells in each area.

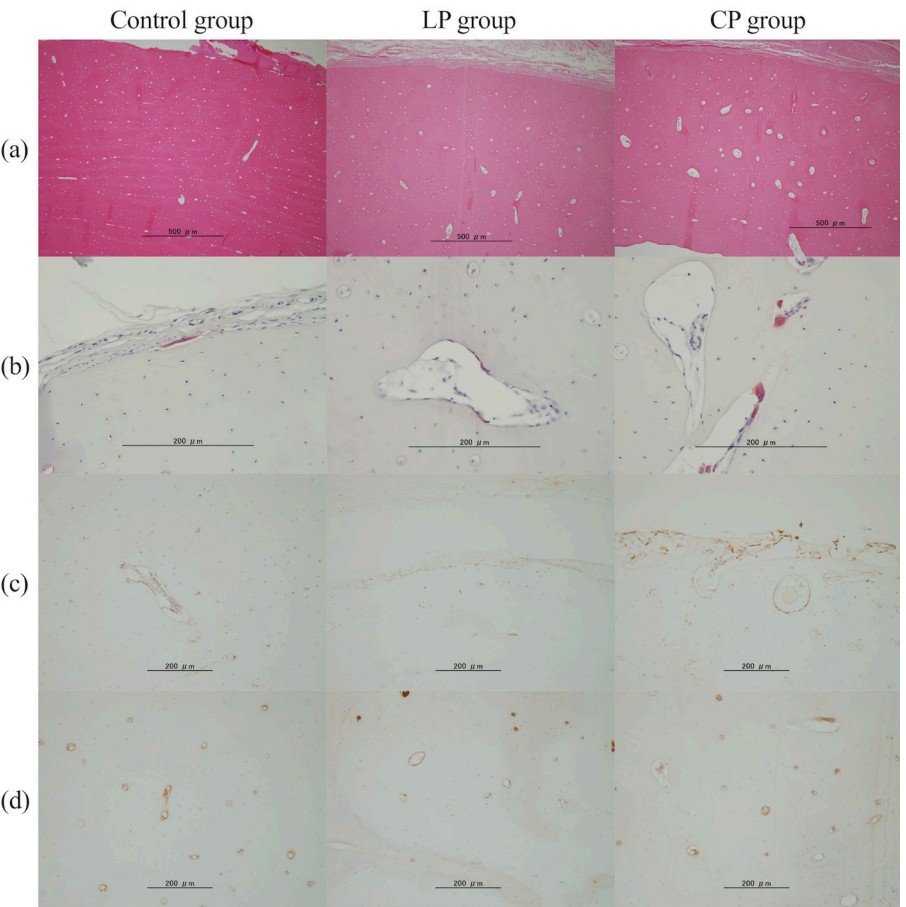

**Fig 5. Each stained image in the cranial area of the radial cortical bone.** a) Hematoxylin and eosin staining. b) Tartrate-resistant acid phosphatase staining. c) RUNX2. d) CD31.

**Table 2. Comparison of area ratios in the transverse radial section.**

|  | Area | Radial cortical area/TA[※2] | Radial marrow cavity area/TA[※2] |
|---|---|---|---|
| Control group (n = 8) | Whole[※1] | 0.75 ± 0.04 | 0.25 ± 0.04 |
|  | Cranial | 0.23 ± 0.02[a] | 0.02 ± 0.01 |
|  | Middle | 0.22 ± 0.03 | 0.2 ± 0.02 |
|  | Caudal | 0.3 ± 0.02 | 0.04 ± 0.02 |
| LP group (n = 3) | Whole[※1] | 0.7 ± 0.05 | 0.3 ± 0.05 |
|  | Cranial | 0.21 ± 0.01 | 0.03 ± 0.02 |
|  | Middle | 0.2 ± 0.03 | 0.23 ± 0.02 |
|  | Caudal | 0.29 ± 0.02 | 0.04 ± 0.01 |
| CP group (n = 3) | Whole[※1] | 0.73 ± 0.05 | 0.27 ± 0.05 |
|  | Cranial | 0.2 ± 0.01[a] | 0.01 ± 0.01 |
|  | Middle | 0.22 ± 0.04 | 0.22 ± 0.03 |
|  | Caudal | 0.32 ± 0.01 | 0.04 ± 0.02 |

[a] p = 0.06, Control group at cranial vs. CP group at cranial

※[1] Total area of the cortex or medullary cavity in the transverse radial section

※[2] Total Area: total area of the transverse section of the radius, including the cortical and medullary cavity.

**Table 3. Comparison of the number of cells using hematoxylin and eosin staining and immunohistochemical staining.**

| | Area※ | Empty lacunae/lacunae (Empty lacunae rate) | TRAP (+) density (NO./mm2) | RUNX2 (+) density (NO./mm2) | CD31 (+) density (NO./mm2) |
|---|---|---|---|---|---|
| Control group (n = 8) | Whole | 0.21 ± 0.07 | 0.12 ± 0.11 | 12.37 ± 6.22 | 4.55 ± 1.24 |
| | Cranial | 0.2 ± 0.07[a] | 0.05 ± 0.08[a] | 15.09 ± 10.9 | 5.47 ± 2.67 |
| | Middle | 0.22 ± 0.06 | 0.07 ± 0.14 | 11.68 ± 5.36 | 3.95 ± 0.84 |
| | Caudal | 0.19 ± 0.09 | 0.2 ± 0.16 | 10.9 ± 5.22 | 4.32 ± 0.95 |
| LP group (n = 3) | Whole | 0.22 ± 0.05 | 0.22 ± 0.2 | 16.33 ± 7.69 | 5.6 ± 1.65 |
| | Cranial | 0.24 ± 0.05[b] | 0.17±0.13[b] | 18.55±9.99 | 5.23±2.11 |
| | Middle | 0.25 ± 0.05 | 0.34 ± 0.4 | 19.06 ± 10.73 | 5.62 ± 1.59 |
| | Caudal | 0.19 ± 0.06 | 0.18 ± 0.21 | 12.47 ± 4.22 | 5.85 ± 1.44 |
| CP group (n = 3) | Whole | 0.33 ± 0.05 | 0.6 ± 0.36 | 20.32 ± 9.01 | 4.09±1.7 |
| | Cranial | 0.39 ± 0.05[a,b] | 1.03 ± 0.57[a,b] | 25.44 ± 9.73 | 4.69 ± 2.59 |
| | Middle | 0.32 ± 0.06 | 0.69 ± 0.4 | 17.06 ± 11.77 | 3.74 ± 2.27 |
| | Caudal | 0.27 ± 0.05 | 0.22 ± 0.15 | 19.05 ± 7.33 | 3.96 ± 0.9 |

[a] $p < 0.05$, Control group at cranial vs. CP group at cranial

[b] $p < 0.05$, LP group at cranial vs. CP group at cranial

※Counting the number of cells in the cortical region only in the radial cross-section

## mRNA expression analysis with qPCR

*Tnfsf11* and *TNF-α* expression was significantly greater in the CP group than in the Control group (p = 0.03 and p = 0.026, respectively). Compared with the Control and LP groups, the CP group showed a significant increase in *IL-6* expression (p = 0.002 and p = 0.017, respectively). Similarly, compared with the other two groups, the CP group showed a significant increase in *HIF-1α* expression (p < 0.001) (Fig 6).

## Discussion

In the field of small animal orthopedics, internal fixation using conventional plates and screws is a useful treatment for radial fractures [19], However, IIO is a common complication of internal fixation using conventional plates and screws, especially in small breed dogs. From this study results, IIO was found to occur in the CP group in which the periosteum was compressed at the time of plate placement. In contrast to the LP group, the CP group exhibited thinning of the cranial cortical bone after 12 weeks, and a reduction in bone mineral density, decrease cortical bone area in the radial cranial area, increase in the osteoclast number, and Tnfsf11 expression in the radius just below the plate after 36 weeks. These results were considered to be a continuation of the effects of cortical osteonecrosis (a remodeling repair process) due to impaired periosteal blood flow after plate placement, which has been reported to occur at 8–12 weeks postoperatively [4]. No similar results were obtained in the LP group. Another possibility was that increased expression of pro-inflammatory cytokines due to the slight friction between the radial surface and the plate (to a certain degree independent of the stiffness of the construct) with a method of compression on the periosteum that occurs during weight-bearing on the forelimb may have enhanced bone resorption.

It has been reported that mechanical stress factors contribute to changes in the cortical bone just below the plate after the application of a conventional plate. The placement of a solid plate with significantly reduced bone strain induced bone formation around the screws proximal to the plate and reduced the stiffness of the bone just below the plate (within 8 weeks of plate placement) [20]. It is important to avoid extensive contact between the implant and bone

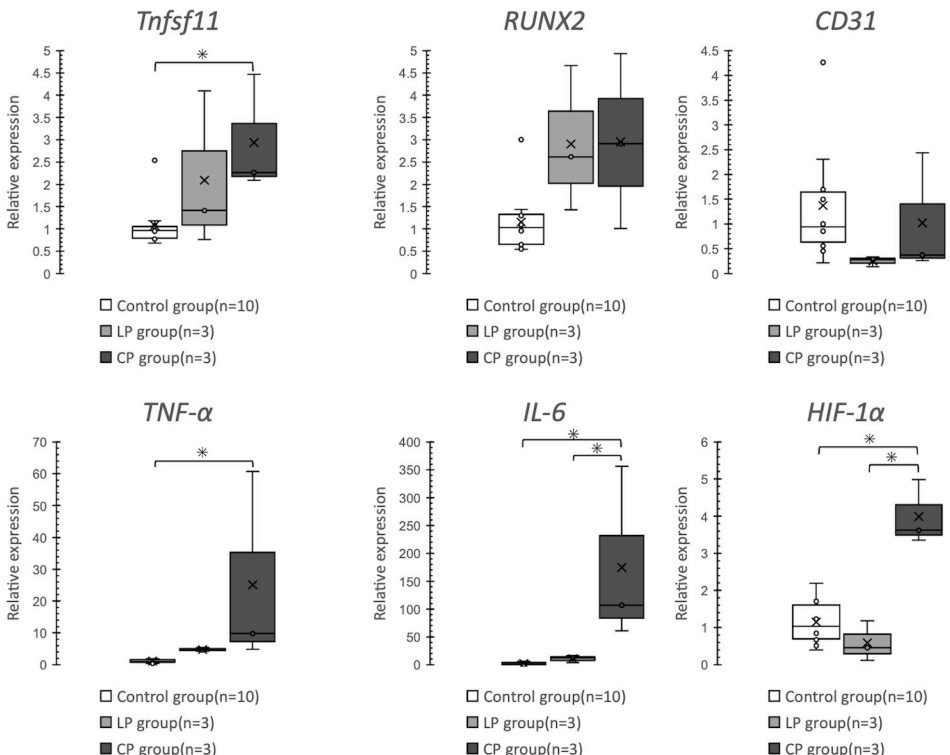

**Fig 6. Comparison of relative mRNA expression levels using the ΔΔCT method.** *Tnfsf11* and *TNF-α* expression is significantly higher in the CP group than in the Control group. *IL-6* and *HIF-1α* expression is significantly higher in the CP group than in the Control and LP groups.

to reduce the influence of biological factors such as impaired blood supply, cortical bone necrosis, or temporary cortical bone porosity [13, 21], In addition, the porosity of the peri-implant cortical bone at 12 weeks after internal fixation has been attributed to impaired blood flow from the extra- and endosseous bone membranes. It has been reported that cortical bone porosity represents an alteration in bone remodeling, remodeling areas are associated with areas of impaired blood flow, and improvement in blood flow circulation reduces bone porosity [22]. There is a negative report on the development of IIO due to mechanical stress in small breed dogs. The comparable normalized stiffness of the radius in small and large dogs (fewer complications after internal fixation relative to those in smaller dogs) with plates suggests that complications in the healing process reported in small dogs may be due to biological, rather than mechanical, factors [23].

In the present study, both CP and LP groups showed bone formation around the screw on the proximal side of the plate on radiographs, suggesting that part of the axial load passed through the plate and the screw, and thus the load on the cortical bone in the center of the underlying plate was reduced to some extent. In the LP group, wherein periosteal compression was avoided, osteoclast numbers and *Tnfsf11* expression, which are bone resorption markers, did not significantly differ from those in the Control group. This result and previous reports suggest that mechanical stress may be not a major cause of IIO occurrence after plate fixation. However, no reports have found biological factors to be the main factor in the development of IIO after 36 weeks of plate and screw placement. Owing to a previous report that attributed

cortical bone porosity within 12 weeks of internal fixation to impaired blood flow, and that areas of impaired blood flow were associated with remodeling areas [22], we considered that it may reflect the persistence of the effects of cortical osteonecrosis due to impaired periosteal blood flow (8–12 weeks post-operatively) and the subsequent reconstructive process that resulted after plate placement as previously reported.

The locking plate method achieves stability between the bone fragments by fixing a screw and plate. In contrast to the conventional plate method, in which stability is achieved by the frictional force between the plate and bone, the locking plate method allows the plate to be placed away from the surface of the periosteum and does not require compression of the periosteum, thus preserving periosteal blood flow and achieving secondary bone healing due to relative stability. Therefore, it is useful as a treatment for fractures in the distal radius of toy breeds, which are considered to have poor soft tissue blood supply [8]. Reducing the area of contact between the bone and the plate is important to maintain the biological activity of the bone and resistance to infection [24], and various studies have reported that the locking plate technique shows superior biological properties than does the conventional plate technique. As such, it has been shown to be effective in the treatment of osteoporotic conditions [25] and infected fractures [26]. In the present study, there was no significant difference in bone mineral density and cortical bone area in the radius just below the plate in the LP and control groups after 36 weeks. The rate of empty lacunae and the number of osteoclasts were significantly lower in the LP group than in the CP group. These findings, as well as the report that the locking plate technique does not induce temporary osteoporosis after surgery [10], suggest that the effect of cortical osteonecrosis due to impaired periosteal blood flow is less pronounced and that the remodeling process is faster. In addition, the difference between the LP and CP groups was with or without compression of the plate to the radial surface, which may be responsible for the difference in results between the groups. These findings support the use of the locking plate technique to preserve periosteal blood flow compared to the conventional plate technique. It may be more useful to use the locking plate technique to reduce cortical osteonecrosis after plate fixation.

The CP group showed a reduction in bone density in QCT, a reduction in cortical bone area, an increase in osteoclast numbers in the radial cranial area by histological examination, and an increase in *Tnfsf11* expression by qPCR. The decreased bone mineral density in the CP group is thought to the consequence of an imbalance between bone formation and resorption due to increased bone resorption. The enhanced bone resorption activity in the CP group may be due to the effects of pro-inflammatory cytokines and hypoxia-inducing factors, based on the increased expression of *TNF-α*, *IL-6*, and *HIF-1α*. Pro-inflammatory cytokines are in the Th1 group, of which *TNF-α* and *IL-1β* stimulate RANKL expression and osteoclast production in bone marrow stromal cells or osteoblasts. *IL-6* interacts with *TNF-α* and *IL-1β* to stimulate osteoclasts and increase the production of RANKL and OPG [16]. Pro-inflammatory cytokines such as *TNF-α*, *IL-1β*, and *IL-6* have been reported to increase osteocyte apoptosis [27], and their direct effects may result in increased osteoclast-driven bone resorption. Pro-inflammatory cytokines such as *TNF-α*, *IL-1β*, and *IL-6* have also been reported to influence bone cells to cause positive feedback and may express greater levels of pro-inflammatory cytokines [28]. In this study, the empty lacunae rate; the number of TRAP-positive osteoclasts; and the expression of *TNF-α*, *IL-6*, and *Tnfsf11* in the CP group were significantly increased relative to those in the Control group. Furthermore, compared with the LP group, the CP group showed significant increase in the empty lacunae rate, the number of TRAP-positive osteoclasts, and *IL-6* expression.

In summary, our results suggest that application of the implant to compress the periosteum may induce apoptosis of osteocytes due to an increase in pro-inflammatory cytokines, which

may promote bone resorption and result in IIO. This phenomenon was observed only in the CP group and not in the LP group. These results suggest that enhanced bone resorption may be due to the persistent effect of cortical osteonecrosis after plate placement, which occurs at 8–12 weeks postoperatively as described above, and due to the increased expression of pro-inflammatory cytokines caused by the slight friction between the radial surface and the plate with a method of compression on the periosteum that occurs during weight-bearing of the forelimb.

In the present model, the differences between the CP and LP groups were the presence or absence of plate compression on the periosteum and the fact that the conventional plate method involved fixation by the friction among the plate, screw, and bone. As periosteal blood flow is associated with blood flow in the lateral one-third of the radial area, compression of the periosteum by the plate may cause impaired perfusion to the cortical bone [29]. Immunostaining and qPCR results for *CD31* in this study did not show a significant difference in the number of vessels between the two groups. These results do not indicate a decrease in the number of vessels in the radial cortical bone when the periosteum is compressed with a plate. Thus, the number of blood vessels may not reflect the actual blood flow to the cortical bone.

The bone loss associated with extra-skeletal diseases such as cancer, rheumatoid arthritis, osteoporosis, fractures, and obstructive pulmonary disease has been suggested to be associated with hypoxia [17]. Monocyte-macrophage cell populations are known to be activated by hypoxia [30]. Hypoxia-induced factor expression in osteoblasts and osteoclasts leads to increased osteoclast formation and mature osteoclast activity due to increased RANKL expression via bone marrow stromal cells or osteoblasts, resulting in enhanced bone resorption. In particular, *HIF-1α* is associated with bone resorption in microenvironmental hypoxia [17] and increases with ischemic osteonecrosis [31]. In addition, activation of *HIF1-α* induces senescence-associated secretory phenotypes (phenomenon in which senescent cells express and secrete a variety of inflammatory proteins into their surroundings) and increases the expression of pro-inflammatory cytokines [32]. In the present study, the CP group showed a significant increase in *HIF-1α* expression in comparison with the Control and LP groups. Since the CP group showed elevated *TNF-α* and *IL-6* expression, it is possible that the plates placed to compress the periosteum in the conventional plate technique induced pro-inflammatory cytokines in the bone just below the plate, resulting in a state of hypoxia in the microenvironment and enhanced bone resorption. Another possibility is that conventional plate compression on the periosteum reduced periosteal blood flow and enhanced bone resorption due to the induction of *TNF-α* and *IL-6* from senescence-associated secretory phenotypes by increased expression of *HIF1-α*.

This study had some limitations. No osteotomy was performed (not an actual fracture model), and the plate was elevated 3 mm from the radial surface in the LP group. Locking systems have been reported to affect biomechanical stiffness when elevated by more than 3 mm, and it is recommended that the gap between the bone and the plate should be less than 2 mm [33]. As an experimental design for this study, we aimed to investigate the long-term (36 weeks) effects of plate compression on the cortical bone just below the plate using a CP group with the plate compressed to the bone surface and an LP group with the plate fixed so that the plate does not contact the bone. Therefore, no osteotomy was performed on the radius and no bridging fixation was performed on the unstable fracture fragments. In the LP group, the space of 3 mm in the present study was established because the priority was to ensure that the plate was not in contact with the bone and that it was close to the clinically recommended 2 mm. The results of this study were not obtained from the clinically recommended (instability exists between bone fragments) use of the locking system. However, they show that the method of without compression of the periosteum is superior to the method of compression of the periosteum. Another study limitation is the use of beagle dogs. It has been recognized that toy breed

dogs are more frequently affected by serious postoperative complications such as osteopenia, re-fracture, or nonunion after radial fracture repair than large breed dogs [2]. Small dogs also have a lower vascular density in the distal radius compared to large dogs, which may contribute to their higher incidence of postoperative complications[5]. Beagle dogs have not been reported to have a higher incidence of radial ulnar fractures or postoperative complications as observed with toy breeds. It is unclear to what extent the results from beagles are applicable to toy breeds, which are common in actual clinical cases. However, it was considered that the results of this study, in which the rocking system had less effect on bone than the conventional system, may be more useful for toy breeds which have less blood flow and a higher incidence of complications.

In summary, application of the plate to compress the periosteal surface was considered to result in increased pro-inflammatory cytokine ($TNF$-$\alpha$, $IL$-$6$) production and hypoxia-inducible factor ($HIF$-$1\alpha$) expression in the bone just below the plate, which promoted bone resorption and induced a reduction in bone mineral density. These results may capture the persistent remodeling process of cortical osteonecrosis due to impaired periosteal blood flow after plate placement which has been reported to occur 8–12 weeks after surgery [4]. Increased expression of pro-inflammatory cytokines due to the slight friction that occurs between the radius and the plate with a method of compression on the periosteum during weight-bearing on the forelimb was also considered to be a factor in promoting bone resorption. This suggests that the locking plate technique is clinically more effective than the conventional plate technique in reducing cortical osteonecrosis caused by impaired periosteal blood flow at 36 weeks after fixation.

## Supporting information

**S1 Checklist.**
(PDF)

## Acknowledgments

We gratefully acknowledge the work of past and present members of our laboratory.

## Author Contributions

**Conceptualization:** Norihiro Muroi, Hiroki Ochi, Masakazu Shimada, Yoshinori Asou, Yasushi Hara.

**Data curation:** Norihiro Muroi.

**Formal analysis:** Norihiro Muroi.

**Investigation:** Norihiro Muroi.

**Methodology:** Norihiro Muroi, Hiroki Ochi, Masakazu Shimada, Yoshinori Asou, Yasushi Hara.

**Project administration:** Norihiro Muroi.

**Writing – original draft:** Norihiro Muroi.

**Writing – review & editing:** Norihiro Muroi, Hiroki Ochi, Masakazu Shimada, Yoshinori Asou, Yasushi Hara.

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
