## [Decision Letter · Decision Letter 0]

21 Dec 2020

PONE-D-20-28976

Effects of long-term plate fixation with different fixation modes on the radial cortical bone in dogs

PLOS ONE

Dear Dr. Muroi,

Thank you for submitting your manuscript to PLOS ONE. After careful consideration, we feel that it has merit but does not fully meet PLOS ONE’s publication criteria as it currently stands. Therefore, we invite you to submit a revised version of the manuscript that addresses the points raised during the review process.

It was indeed a difficult task to find reviewers. I apologize for any inconvenience it may have caused. 14 invited reviewers declined, 4 agreed, however, they did not complete their job. Finally 2 reviewers made the review.

The reviewes comments will help to improve your study.

Important: Pleas seek help from a native speaker to improve your english.

We look forward to receiving your revised manuscript.

Kind regards,

Hans-Peter Simmen, M.D., Professor of Surgery

Academic Editor

PLOS ONE

Journal Requirements:

2.) In your Methods section, please provide additional details regarding the dogs used in your study and ensure you have described the source. For more information regarding PLOS' policy on materials sharing and reporting, see https://journals.plos.org/plosone/s/materials-and-software-sharing#loc-sharing-materials.

3.) Please include captions for your Supporting Information files at the end of your manuscript, and update any in-text citations to match accordingly. Please see our Supporting Information guidelines for more information: http://journals.plos.org/plosone/s/supporting-information.

Reviewers' comments:

Reviewer's Responses to Questions

**Comments to the Author**

1. Is the manuscript technically sound, and do the data support the conclusions?

Reviewer #1: Yes

Reviewer #2: Partly

2. Has the statistical analysis been performed appropriately and rigorously? 

Reviewer #1: I Don't Know

Reviewer #2: No

3. Have the authors made all data underlying the findings in their manuscript fully available?

Reviewer #1: No

Reviewer #2: Yes

4. Is the manuscript presented in an intelligible fashion and written in standard English?

Reviewer #1: No

Reviewer #2: Yes

5. Review Comments to the Author

Reviewer #1: General comments:

Please explain AND references why you chose the specific stains that you did for this paper.

Please discuss in your discussion the differences between research beagles and toy breed dogs where locking plates are most useful.

Specific comments:

Line 56-58: This is a run-on sentence. Please divide into two sentences or change the wording in some way.

Line 59: Should you say locking plates placed in a low contact method or Low contact locking plates? Some surgeons still place these locking plates directly on the periosteum.

Line 60: There is another older reference for this point that you should consider adding.

Appendicular fracture repair in dogs using the locking compression plate system: 47 cases.

Haaland PJ, Sjöström L, Devor M, Haug A.

Vet Comp Orthop Traumatol. 2009;22(4):309-15. doi: 10.3415/VCOT08-05-0044. Epub 2009 Jun 23.

PMID: 19597631

Line 65: End this sentence after "healing" and start a second sentence with "Appropriate blood flow reduces the risk of infection and IIO".

Line 66-67: Please make this into a complete sentence.

Line 70: This is not true. Here is a clinical paper in vet med that compares clinical outcomes between locking plates and Dynamic compression plates.

Outcome of Repair of Distal Radial and Ulnar Fractures in Dogs Weighing 4 kg or Less Using a 1.5-mm Locking Adaption Plate or 2.0-mm Limited Contact Dynamic Compression Plate.

Nelson TA, Strom A.

Vet Comp Orthop Traumatol. 2017 Nov;30(6):444-452. doi: 10.3415/VCOT-17-01-0005. Epub 2017 Dec 4.

PMID: 29202508

Line 81: CT scans of what? Just the radii?

Line 85: please add "in either group" if that is true.

Line 107: I think "contouring" is a better word.

Line 113: Please briefly describe the other screws that were placed. All holes filled with locking screws?

Line 116: same as above. Just briefly describe the placement of the rest of the screws.

Line 130: What was the method to determine bone density on a radiograph?

Line 237: in how many patients?

Line 240: Pain on palpation or just lameness? Be specific.

Line 245: say "in either group"

Line 248: who made these determinations? a surgeon or a radiologist?

Line 250-1: in which group?

Line 324: consider adding "especially in small breed dogs"

Line 324-329: run-on sentence. Please fix

Line 329: I would move this sentence to after the next sentence and I would remove everything after "group".

Line 330: Replace "This" for "These" as you describe many results.

Line 337-41: Again this sentence is unclear because it is too long. Please adjust.

Line 341: Remove "In contrast"

Line 343-48: Sentence too long.

Line 348: what does negative report mean in this context?

Line 349: please re-phrase "plate-placed"

Reviewer #2: PONE-D-20-28976

The goal of the author(s) was, to compare the behaviour of bone with plates fixed onto the intact radial bones of dogs (n=3). The plates were fixed either in a fashion which applies pressure onto the periosteum and the bone or in a fashion, which avoids compression. The study lasted for 36 weeks. The outcome was observed by radiographic examination, microbiology and quantitative polymerase chain reaction analysis.

The manuscript reviewed is well structured and easy to read. The reviewer points out a few aspects in the terminology used, the statistics involved and the conclusions made which could help to further improve the manuscript.

The review is mostly based on a consecutive order where minor and mayor findings are mixed and happen to be discussed when they appeared in the manuscript. Line numbers as they are used in the manuscript helped to structure the review. On certain places, reference was made to line numbers further down in the manuscript.

The reviewer prefers to explain the findings he made or the flaws he might have found and gives examples of how he came to the conclusions drawn.

19 (see 230 as well) The abbreviation of the two plate fixation methods was based by the author(s) on the screw used to fix the plates, locking screw against cortex screw. In fact, both screws grab the cortex to fix the plate (where the locking screw, in addition, also grabs the plate). Based on this, it might be better to base the abbreviation on the mechanics involved. In LP: the L could stand for locking and in CP the C could stand for compression (or compressing) rather than for cortex. Therefore, rephrasing of 19, 20 to ...locking head screw (Locking Plate group, LP) and 21 to ...a cortex screw (Compression plate group, CP) would be encouraged by the reviewer as this involves no need to change the abbreviations used by the author(s) in the manuscript but points to the crucial difference in the fixation of the plate used, which is the same in both procedures, an LCP.

63 (see 371 also) ...preserving periosteal blood flow and biological healing with callus formation. In the reviewers opinion, bone healing is represented by a reconstitution of the ability of the (broken) bone, to restore the stability aka stiffness of the bone, or the extremity, to a state, prior to the accident. The course which leads there, could follow a route with or without the formation of callus and leading to the same result and is, in any case, biological. In fact, LP might more often lead to healing without callus than with callus and this fact could be helpful in the treatment of forearm fractures because too much callus could inhibit proper function of the forearm.

107 LCP contouring instead of LCP countering (?)

137 ...center of the plate: midway between the 3rd and 4th hole screws. The center of a nine hole plate (in the long axis) appears to be hole number five from either end but not between hole number three and four. It was only when the reviewer observed Fig. 1 that he realised that, of the nine hole plate, six holes only were occupied by screws. There was no statement found in the manuscript regarding that fact. Omitting hole in the sentence above ...the 3rd and 4th hole screws... might clarify the situation (in addition to a paragraph in the text which states, that the three middle holes of the plate were not occupied).

145 The reviewer did not understood the term harvested in segmental fashion as, after harvesting, the bone was divided into segments (see next sentence) and thus, segmented after harvesting where, during harvesting, the bone was still intact (?)

157, 158 washed with 90% ethanol and then with 99.5% and then dehydrated with 99.5% what is the difference between the washing and dehydrating? Duration of exposure? If so, or other, please state.

175 ...phosphorus A mixture... or ...phosphorus. A mixture... ?

228 The number of specimens used is small (n=3). In that case the reviewer strongly advises to use descriptive statistics in favour of inferential statistics. Inferential statistics might be reasonable where the number of observations is high (>=10) and the distribution of samples can be proven to be normally distributed. Data, which do not represent this criteria(s) (which, in biological tests with small samples is more often the case than not) might be presented as box-plots rather than bar-graphs with the benefit, that the reader can observe on his or her own, where the data trend to and if the observations are reasonable at all.

230,231 (19,20,21 also) The abbreviation of the two plate fixation methods was based by the author(s) on the screw used to fix the plates, locking screw against cortex screw. In fact, both screws grab the cortex to fix the plate (where the locking screw, in addition, also grabs the plate). Based on this, it might be better to base the abbreviation on the mechanics involved. In LP: the L could stand for locking and in CP the C could stand for compression (or compressing) rather than for cortex. Therefore, rephrasing of 230,231 to ...locking head screws was defined as the LP (Locking Plate) group, and the group in which LCPs were fixed with cortex screws was defined as the CP (Compression Plate) group. would be encouraged by the reviewer as this involves no need to change the abbreviations used by the author(s) in the manuscript but points to the crucial difference in the fixation of the LCP used.

249 There is an inconsistency in the use of terms in case the timing; weeks as well as months are used. The reviewer advises to use weeks in the manuscript all over and to change the months into weeks so that 249 reads ...after 12 weeks. and 256 reads ...12 weeks 24 weeks and 36 weeks after... (see also 277).

265 (also 263) By definition there is no point in time which goes beyond 36 weeks of observation as the animals were killed after 36 weeks. Therefore, the term after should be used carefully and should exclude situations, where the reader might interprete it as beyond 36 weeks. The author(s) might try to say, that the density significantly decreased during the observation period. However, the manuscript states, that 265 ...mineral density significantly decreased after 36 weeks of implantation. To change the after into during might be advisable.

268 ...and after 36 weeks. Proposal, change to: ...and 36 weeks after surgery.

277 change 9 months into 36 weeks

287-290 There is no need to present a scaling factor (x100 and so) as, with one exception (left image of Fig 2) the images contain rulers which are more universal than scaling factors.

335 The reviewer did not understand this sentence as it could have different meanings. If, however, the author(s) try to state, that the modulus aka stiffness of the plate has an important role, the reviewer has a different opinion: If, in this setup, friction happens between plate and bone it is likely that the modulus of the plate (or the stiffness of the construct) plays a minor role. The stiffness of the CP might be inferior to that of the LP but still in a range, which guarantees a correct fixation of a, although here not present, fracture. It is more likely, that the fact in itself, that the CP could glide on the periosteum (to a certain degree independent of the stiffness of the construct) could lead to inflammation.

344 change ...three months into ...12 weeks

371 ...preserving periosteal blood flow and biological healing with callus formation. In the reviewers opinion, bone healing is represented by a reconstitution of the ability of the (broken) bone, to restore the stability aka stiffness of the bone, or the extremity, to a state, prior to the accident. The course which leads there, could follow a route with or without the formation of callus and leading to the same result and is, in any case, biological. In fact, LP might more often lead to healing without callus than with callus and this fact could be helpful in the treatment of forearm fractures because too much callus could inhibit proper function of the forearm.

387 ...use of the locking plate technique to promote biological healing in comparison with the conventional plate technique. As stated above, bone healing is biological and the three corners are: blood supply, stability and biology. Both plate constructs lead to healing by biological processes, in that respect, they can‘t be compared. However, the LP has a benefit when it comes to the preservation of blood supply and that fact might be compared.

415 see 335

423 In the opinion of the reviewer, the number of vessels are not necessarily relevant if there is no proof, that the vessels still can play there role and help to perfuse the bone. However, at 429-447 the author(s) discussed a different route, which is more in favour of the reviewers opinion.

424 The reviewer has difficulties to understand the terminology. What does the author(s) try to explain with These results do not clearly indicate that the IIO in this study was caused by biological factors. Changes in a biological system happen to be biologic. If this is not the case here, what are the factors then, in the authors opinion?

457 Increasing the spacing of the LP can also increase the cross sectional area and thus the stiffness of the construct. This procedure could be beneficial in certain load cases and presents, for the reviewer, not a drawback as such. The surgeons using this plate in a clinical setting might prefer to use it according to SOP and thus without a spacer. If properly applied, the system in it self elevates the plate from the bone, when the screws are tightened. The space created is enough so that the periosteum is not compressed. However, the fact that the author(s) did not fix the plate in that manner could lead to the assumption, that the results observed could be solely related to this, non conformal application of the plate.

466 see 423

470 see 335

Fig 4 and Fig 6 The results presented here in bar graph form augment (in the reviewers opinion in a negative way) what was said under 228 above. To present the findings in box-plots instead of bar-graphs would greatly increase the readability of the results found and would help to distinguish between trends and/or significances, if any can be found.

6. PLOS authors have the option to publish the peer review history of their article (what does this mean?). If published, this will include your full peer review and any attached files.

Reviewer #1: No

Reviewer #2: **Yes: **Urs Schlegel

---

## [Author Response · Author response to Decision Letter 0]

2 Feb 2021

Manuscript ID: PONE-D-20-28976

Title: Effects of long-term plate fixation with different fixation modes on the radial cortical bone in dogs

Point-by-Point Response

Thank you for your review of our paper. We have addressed each of your concerns below. Revisions in the text are highlighted in red font and the changes are indicated by the subscription Line numbers.

Response to Reviewer 1 

1) Please explain AND references why you chose the specific stains that you did for this paper.

Response: We have added these details in the “Introduction” (p. 5–6, lines 73–87).

2) Please discuss in your discussion the differences between research beagles and toy breed dogs where locking plates are most useful. 

Response: We have modified the “Discussion” accordingly (p. 34, lines 480–490).

3) Line 56-58: This is a run-on sentence. Please divide into two sentences or change the wording in some way.

Response: We have revised this sentence (p. 4, lines 55–58).

4) Line 59: Should you say locking plates placed in a low contact method or Low contact locking plates? Some surgeons still place these locking plates directly on the periosteum.

Response: Although a correlation between plate contact area and area of osteonecrosis has been reported in dogs (Lippuner K et al, Arch Orthop Trauma Surg. 1992), there have been contradictory reports with no difference in the vascularisation and porosity between DCPs and LC-DCPs in canine fracture models (Jain R et al, J Trauma. 1998). The contact area between the plate and the bone is considered to be influenced not only by the undersurface of the plate, but also by the shape of the bone surface. In addition, although the LCP can be placed away from the bone surface using the LHS, it is reported that the LCP may be in contact with the bone when it is actually placed (Ahmad M et al, Injury. 2007). From the above, I should say locking plates placed in a low contact method.

5) Line 60: There is another older reference for this point that you should consider adding. 

Appendicular fracture repair in dogs using the locking compression plate system: 47 cases.

Haaland PJ, Sjöström L, Devor M, Haug A.

Vet Comp Orthop Traumatol. 2009;22(4):309-15. doi: 10.3415/VCOT08-05-0044. Epub 2009 Jun 23. PMID: 19597631 

Response: Thank you for your suggestion. We have added a new reference #9.

6) Line 65: End this sentence after "healing" and start a second sentence with "Appropriate blood flow reduces the risk of infection and IIO".

Response: We have revised the text accordingly (p. 4–5, lines 64–67).

7) Line 66-67: Please make this into a complete sentence.

Response: We have revised the text accordingly (p. 5, lines 66–67).

8) Line 70: This is not true. Here is a clinical paper in vet med that compares clinical outcomes between locking plates and Dynamic compression plates. 

Outcome of Repair of Distal Radial and Ulnar Fractures in Dogs Weighing 4 kg or Less Using a 1.5-mm Locking Adaption Plate or 2.0-mm Limited Contact Dynamic Compression Plate.

Nelson TA, Strom A.

Vet Comp Orthop Traumatol. 2017 Nov;30(6):444-452. doi: 10.3415/VCOT-17-01-0005. Epub 2017 Dec 4. 

PMID: 29202508 

Response: Thank you for providing these insights. We apologize for the incorrect expression of the sentence. We have revised the text accordingly (p. 5, line 68).

9) Line 81: CT scans of what? Just the radii?

Response: We have added a new sentence (p. 6–7, lines 98–99).

10) Line 85: please add "in either group" if that is true.

Response: We have revised the text accordingly (p. 7, lines 102–103).

11) Line 107: I think "contouring" is a better word.

Response: We have revised to use the word “contouring" (p. 8, lines 124)

12) Line 113: Please briefly describe the other screws that were placed. All holes filled with locking screws? 

Response: We have provided additional details (p. 8–9, lines 130–132).

13) Line 116: same as above. Just briefly describe the placement of the rest of the screws.

Response: We have added a new sentence (p. 9, lines 135–137).

14) Line 130: What was the method to determine bone density on a radiograph?

Response: The bone density on a radiograph was subjectively assessed.

15) Line 237: in how many patients?

Response: We have added the number of patients (p. 18, line 257).

16) Line 240: Pain on palpation or just lameness? Be specific.

Response: We have revised the sentence for clarity (p. 18, lines 258–259).

17) Line 245: say "in either group" 

Response: We have revised the text accordingly (p. 19, lines 264–265).

18) Line 248: who made these determinations? a surgeon or a radiologist? 

Response: The radiographic observations were all made by a surgeon.

19) Line 250-1: in which group?

Response: We have provided these details (p. 19, lines 270).

20) Line 324: consider adding "especially in small breed dogs"

Response: We have revised the text accordingly (p. 26, lines 343).

21) Line 324-329: run-on sentence. Please fix 

Response: We have revised the sentence to enhance clarity (p. 26, lines 344–348).

22) Line 329: I would move this sentence to after the next sentence and I would remove everything after "group".

Response: We have revised the text accordingly (p. 26, lines 348–352).

23) Line 330: Replace "This" for "These" as you describe many results.

Response: We have revised the text accordingly (p. 26, lines 348).

24) Line 337-41: Again this sentence is unclear because it is too long. Please adjust.

Response: We have revised the sentence to enhance clarity (p. 27, lines 357–361).

25) Line 341: Remove "In contrast"

Response: We have removed the words “In contrast” (p. 27, lines 361).

26) Line 343-48: Sentence too long.

Response: We have revised the text accordingly (p. 27, lines 363–372).

27) Line 348: what does negative report mean in this context? 

Response: A negative report on the development of IIO due to mechanical stress was indicated by reference “Gauthier CM, Conrad BP, Lewis DD, Pozzi A. In vitro comparison of stiffness of plate fixation of radii from large- and small-breed dogs. Am J Vet Res. 2011;72(8):1112-7.”. We have revised the text to enhance clarity (p. 27, lines 368–372).

28) Line 349: please re-phrase "plate-placed" 

Response: We have revised the text accordingly (p. 27, lines 369–372).

Response to Reviewer 2 

The goal of the author(s) was, to compare the behaviour of bone with plates fixed onto the intact radial bones of dogs (n=3). The plates were fixed either in a fashion which applies pressure onto the periosteum and the bone or in a fashion, which avoids compression. The study lasted for 36 weeks. The outcome was observed by radiographic examination, microbiology and quantitative polymerase chain reaction analysis. 

The manuscript reviewed is well structured and easy to read. The reviewer points out a few aspects in the terminology used, the statistics involved and the conclusions made which could help to further improve the manuscript.

The review is mostly based on a consecutive order where minor and mayor findings are mixed and happen to be discussed when they appeared in the manuscript. Line numbers as they are used in the manuscript helped to structure the review. On certain places, reference was made to line numbers further down in the manuscript. 

The reviewer prefers to explain the findings he made or the flaws he might have found and gives examples of how he came to the conclusions drawn. 

Response: Thank you for reviewing this paper. We have incorporated your suggestions throughout our paper.

1) 19 (see 230 as well) The abbreviation of the two plate fixation methods was based by the author(s) on the screw used to fix the plates, locking screw against cortex screw. In fact, both screws grab the cortex to fix the plate (where the locking screw, in addition, also grabs the plate). Based on this, it might be better to base the abbreviation on the mechanics involved. In LP: the L could stand for locking and in CP the C could stand for compression (or compressing) rather than for cortex. Therefore, rephrasing of 19, 20 to ...locking head screw (Locking Plate group, LP) and 21 to ...a cortex screw (Compression plate group, CP) would be encouraged by the reviewer as this involves no need to change the abbreviations used by the author(s) in the manuscript but points to the crucial difference in the fixation of the plate used, which is the same in both procedures, an LCP. 

Response: We have revised the sentence to enhance clarity (p. 2, lines 19–21).

2) 63 (see 371 also) ...preserving periosteal blood flow and biological healing with callus formation. In the reviewers opinion, bone healing is represented by a reconstitution of the ability of the (broken) bone, to restore the stability aka stiffness of the bone, or the extremity, to a state, prior to the accident. The course which leads there, could follow a route with or without the formation of callus and leading to the same result and is, in any case, biological. In fact, LP might more often lead to healing without callus than with callus and this fact could be helpful in the treatment of forearm fractures because too much callus could inhibit proper function of the forearm.

Response: We have rewritten this sentence accordingly (p. 5, line 64)

3) 107 LCP contouring instead of LCP countering (?)

Response: We apologize for this inadvertent error. We have corrected the word to “contouring" (p. 8, lines 124)

4) 137 ...center of the plate: midway between the 3rd and 4th hole screws. The center of a nine hole plate (in the long axis) appears to be hole number five from either end but not between hole number three and four. It was only when the reviewer observed Fig. 1 that he realised that, of the nine hole plate, six holes only were occupied by screws. There was no statement found in the manuscript regarding that fact. Omitting hole in the sentence above ...the 3rd and 4th hole screws... might clarify the situation (in addition to a paragraph in the text which states, that the three middle holes of the plate were not occupied).

Response: We have removed the word “hole" (p. 10, lines 158–159) and added a new sentence (p. 8–9, lines 130–132 and lines 135–137)

5) 145 The reviewer did not understood the term harvested in segmental fashion as, after harvesting, the bone was divided into segments (see next sentence) and thus, segmented after harvesting where, during harvesting, the bone was still intact (?)

Response: We have revised the sentence to enhance clarity (p. 11, lines 166–167).

6) 157, 158 washed with 90% ethanol and then with 99.5% and then dehydrated with 99.5% what is the difference between the washing and dehydrating? Duration of exposure? If so, or other, please state.

Response: We have revised the sentence (p. 11, lines 177–178).

7) 175 ...phosphorus A mixture... or ...phosphorus. A mixture... ?

Response: We have revised the sentence accordingly (p. 12, line 196)

8) 228 The number of specimens used is small (n=3). In that case the reviewer strongly advises to use descriptive statistics in favour of inferential statistics. Inferential statistics might be reasonable where the number of observations is high (>=10) and the distribution of samples can be proven to be normally distributed. Data, which do not represent this criteria(s) (which, in biological tests with small samples is more often the case than not) might be presented as box-plots rather than bar-graphs with the benefit, that the reader can observe on his or her own, where the data trend to and if the observations are reasonable at all.

Response: We have revised Figs. 4 and 6 from bar-graphs to box-plots.

9) 230,231 (19,20,21 also) The abbreviation of the two plate fixation methods was based by the author(s) on the screw used to fix the plates, locking screw against cortex screw. In fact, both screws grab the cortex to fix the plate (where the locking screw, in addition, also grabs the plate). Based on this, it might be better to base the abbreviation on the mechanics involved. In LP: the L could stand for locking and in CP the C could stand for compression (or compressing) rather than for cortex. Therefore, rephrasing of 230,231 to ...locking head screws was defined as the LP (Locking Plate) group, and the group in which LCPs were fixed with cortex screws was defined as the CP (Compression Plate) group. would be encouraged by the reviewer as this involves no need to change the abbreviations used by the author(s) in the manuscript but points to the crucial difference in the fixation of the LCP used.

Response: We have revised the sentence to enhance clarity (p. 17–18, lines 249–251).

10) 249 There is an inconsistency in the use of terms in case the timing; weeks as well as months are used. The reviewer advises to use weeks in the manuscript all over and to change the months into weeks so that 249 reads ...after 12 weeks. and 256 reads ...12 weeks 24 weeks and 36 weeks after... (see also 277).

Response: We have revised the word from “months" to “weeks" throughout the paper.

11) 265 (also 263) By definition there is no point in time which goes beyond 36 weeks of observation as the animals were killed after 36 weeks. Therefore, the term after should be used carefully and should exclude situations, where the reader might interprete it as beyond 36 weeks. The author(s) might try to say, that the density significantly decreased during the observation period. However, the manuscript states, that 265 ...mineral density significantly decreased after 36 weeks of implantation. To change the after into during might be advisable. 

Response: We have revised the word from “after" to “during" (p. 19–20, lines 281–284)

12) 268 ...and after 36 weeks. Proposal, change to: ...and 36 weeks after surgery.

Response: We have revised the sentence accordingly (p. 20, line 286).

13) 277 change 9 months into 36 weeks

Response: We have revised the text accordingly (p. 21, line 295).

14) 287-290 There is no need to present a scaling factor (x100 and so) as, with one exception (left image of Fig 2) the images contain rulers which are more universal than scaling factors.

Response: We have removed the scaling factor (p. 21, lines 306–309).

15) 335 The reviewer did not understand this sentence as it could have different meanings. If, however, the author(s) try to state, that the modulus aka stiffness of the plate has an important role, the reviewer has a different opinion: If, in this setup, friction happens between plate and bone it is likely that the modulus of the plate (or the stiffness of the construct) plays a minor role. The stiffness of the CP might be inferior to that of the LP but still in a range, which guarantees a correct fixation of a, although here not present, fracture. It is more likely, that the fact in itself, that the CP could glide on the periosteum (to a certain degree independent of the stiffness of the construct) could lead to inflammation.

Response: We have the sentence to enhance clarity (p. 26, lines 354–355).

16) 344 change ...three months into ...12 weeks

Response: We have revised the text accordingly (p. 27, line 364).

17) 371 ...preserving periosteal blood flow and biological healing with callus formation. In the reviewers opinion, bone healing is represented by a reconstitution of the ability of the (broken) bone, to restore the stability aka stiffness of the bone, or the extremity, to a state, prior to the accident. The course which leads there, could follow a route with or without the formation of callus and leading to the same result and is, in any case, biological. In fact, LP might more often lead to healing without callus than with callus and this fact could be helpful in the treatment of forearm fractures because too much callus could inhibit proper function of the forearm.

Response: We have rewritten the text in accord with your comments (p. 29, lines 391–392).

18) 387 ...use of the locking plate technique to promote biological healing in comparison with the conventional plate technique. As stated above, bone healing is biological and the three corners are: blood supply, stability and biology. Both plate constructs lead to healing by biological processes, in that respect, they can‘t be compared. However, the LP has a benefit when it comes to the preservation of blood supply and that fact might be compared.

Response: We have revised the sentence accordingly (p. 30, lines 407–408)

19) 415 see 335

Response: We have sentence to enhance clarity (p. 31, lines 436).

20) 423 In the opinion of the reviewer, the number of vessels are not necessarily relevant if there is no proof, that the vessels still can play there role and help to perfuse the bone. However, at 429-447 the author(s) discussed a different route, which is more in favour of the reviewers opinion.

Response: We have rewritten the sentence accordingly (p. 32, lines 444–447).

21) 424 The reviewer has difficulties to understand the terminology. What does the author(s) try to explain with These results do not clearly indicate that the IIO in this study was caused by biological factors. Changes in a biological system happen to be biologic. If this is not the case here, what are the factors then, in the authors opinion?

Response: We have revised the sentence to enhance clarity (p. 32, lines 444–447).

22) 457 Increasing the spacing of the LP can also increase the cross sectional area and thus the stiffness of the construct. This procedure could be beneficial in certain load cases and presents, for the reviewer, not a drawback as such. The surgeons using this plate in a clinical setting might prefer to use it according to SOP and thus without a spacer. If properly applied, the system in it self elevates the plate from the bone, when the screws are tightened. The space created is enough so that the periosteum is not compressed. However, the fact that the author(s) did not fix the plate in that manner could lead to the assumption, that the results observed could be solely related to this, non conformal application of the plate.

Response: Although the results of this study were not obtained from the clinically recommended use of the locking system (instability exists between bone fragments), we considered our results to indicate that the method without compression of the periosteum was superior to the comparably method with compression. We have revised the text accordingly (p. 34, lines 478–481).

23) 466 see 423

Response: We have revised the sentence to enhance clarity (p. 35, lines 495–497).

24) 470 see 335 

Response: We have revised the sentence to enhance clarity (p. 35, line 499).

25) Fig 4 and Fig 6 The results presented here in bar graph form augment (in the reviewers opinion in a negative way) what was said under 228 above. To present the findings in box-plots instead of bar-graphs would greatly increase the readability of the results found and would help to distinguish between trends and/or significances, if any can be found. 

Response: We have changed Figs. 4 and 6 from bar-graphs to box-plots.

---

## [Editor Report · Decision Letter 1]

8 Feb 2021

Effects of long-term plate fixation with different fixation modes on the radial cortical bone in dogs

PONE-D-20-28976R1

Dear Dr. Muroi,

We’re pleased to inform you that your manuscript has been judged scientifically suitable for publication and will be formally accepted for publication once it meets all outstanding technical requirements.

Kind regards,

Hans-Peter Simmen, M.D., Professor of Surgery

Academic Editor

PLOS ONE
---

## [Editor Report · Acceptance letter]

10 Feb 2021

PONE-D-20-28976R1 

Effects of long-term plate fixation with different fixation modes on the radial cortical bone in dogs 

Dear Dr. Muroi:

I'm pleased to inform you that your manuscript has been deemed suitable for publication in PLOS ONE. Congratulations! Your manuscript is now with our production department. 

Kind regards, 

on behalf of

Dr. Hans-Peter Simmen 

Academic Editor

PLOS ONE